# Nutrient dominance governs the assembly of microbial communities in mixed nutrient environments

**Sylvie Estrela[1†]\*, Alicia Sanchez-Gorostiaga[1,2†], Jean CC Vila[1†], Alvaro Sanchez[1]\***

[1]Department of Ecology & Evolutionary Biology and Microbial Sciences Institute, Yale University, New Haven, United States; [2]Department of Microbial Biotechnology, Centro Nacional de Biotecnología, CSIC, Cantoblanco, Madrid, Spain

**Abstract** A major open question in microbial community ecology is whether we can predict how the components of a diet collectively determine the taxonomic composition of microbial communities. Motivated by this challenge, we investigate whether communities assembled in pairs of nutrients can be predicted from those assembled in every single nutrient alone. We find that although the null, naturally additive model generally predicts well the family-level community composition, there exist systematic deviations from the additive predictions that reflect generic patterns of nutrient dominance at the family level. Pairs of more-similar nutrients (e.g. two sugars) are on average more additive than pairs of more dissimilar nutrients (one sugar–one organic acid). Furthermore, sugar–acid communities are generally more similar to the sugar than the acid community, which may be explained by family-level asymmetries in nutrient benefits. Overall, our results suggest that regularities in how nutrients interact may help predict community responses to dietary changes.

**\*For correspondence:**
sylvie.estrela@yale.edu (SE);
alvaro.sanchez@yale.edu (AS)

[†]These authors contributed equally to this work

**Competing interests:** The authors declare that no competing interests exist.

## Introduction

Understanding how the components of a complex biological system combine to produce the system's properties and functions is a fundamental question in biology. Answering this question is central to solving many fundamental and applied problems, such as how multiple genes combine to give rise to complex traits (*Phillips, 2008*; *Mackay, 2014*), how multiple drugs affect the evolution of resistance in bacteria and cancer cells (*Michel et al., 2008*; *Wood et al., 2014*), how multiple environmental stressors affect bacterial physiology (*Cruz-Loya et al., 2019*), or how multiple species affect the function of a microbial consortium (*Sanchez-Gorostiaga et al., 2019*; *Gould et al., 2018*; *Guo and Boedicker, 2016*).

In microbial population biology, a major related open question is whether we can predict how the components of a diet collectively determine the taxonomic and functional composition of microbial communities. Faith and co-workers tackled this question using a defined gut microbial community and a host diet with varying combinations of four macronutrients (*Faith et al., 2011*). This study found that community composition in combinatorial diets could be predicted from communities assembled in separate nutrients using an additive linear model. Given the presence of a host and its own possible interactions with the nutrients and resident species, it is not immediately clear whether such additivity is directly mediated by interactions between the community members and the supplied nutrients or whether it is mediated by the host, for instance by producing additional nutrients, or through potential interactions between its immune system and the community members. More recently, *Enke et al., 2019* found evidence that marine communities assembled in mixes of two

different polysaccharides could be explained as a linear combination of the communities assembled in each polysaccharide in isolation.

Despite the important insights provided by both of these studies, we do not yet have a general quantitative understanding of how specific nutrients combine together to shape the composition of self-assembled communities (*Pacheco and Segrè, 2020*). Motivated by this challenge, here we use an enrichment community approach (i.e. where natural microbial communities are grown in a defined growth medium under well-controlled lab conditions) to systematically investigate whether the assembly of enrichment microbial communities in a collection of defined nutrient mixes could be predicted from the communities that assembled in each of the single nutrients in isolation.

## Results

### A null expectation for community assembly in mixed nutrient environments

To investigate whether communities assembled in pairs of nutrients can be predicted from those assembled in every single nutrient alone, we must first develop a quantitative null model that predicts community composition in a mixed nutrient environment in the case where each nutrient recruits species independently. Any deviation between the null model prediction and the observed (measured) composition reveals that nutrients are not acting independently, but rather 'interact' to shape community composition. This definition of an interaction as a deviation from a null model that assumes independent effects is commonplace in systems-level biology (*Sanchez, 2019*; *Tekin et al., 2018*).

In order to formulate the null expectation for independently acting nutrients, let us consider a simple environment consisting of two unconnected demes where two bacterial species, A and B, can grow together. The first deme contains a single growth limiting nutrient (nutrient 1), while the second deme contains a different single limiting nutrient (nutrient 2) (*Figure 1A*). In this scenario, each nutrient influences the abundance of species A and B independently: the microbes growing on nutrient one do not have access to nutrient two and vice versa. Let's denote the abundance of species A in demes 1 and 2 by $n_{A,1}$ and $n_{A,2}$, and the abundance of species B as $n_{B,1}$ and $n_{B,2}$, respectively. If we now consider the two-deme environment as a whole, the abundance of species A is the sum of its abundance in each deme $n_{A,12} = n_{A,1} + n_{A,2}$ (likewise, for species B $n_{B,12} = n_{B,1} + n_{B,2}$). This example illustrates that in the scenario when two limiting nutrients act independently, each of them

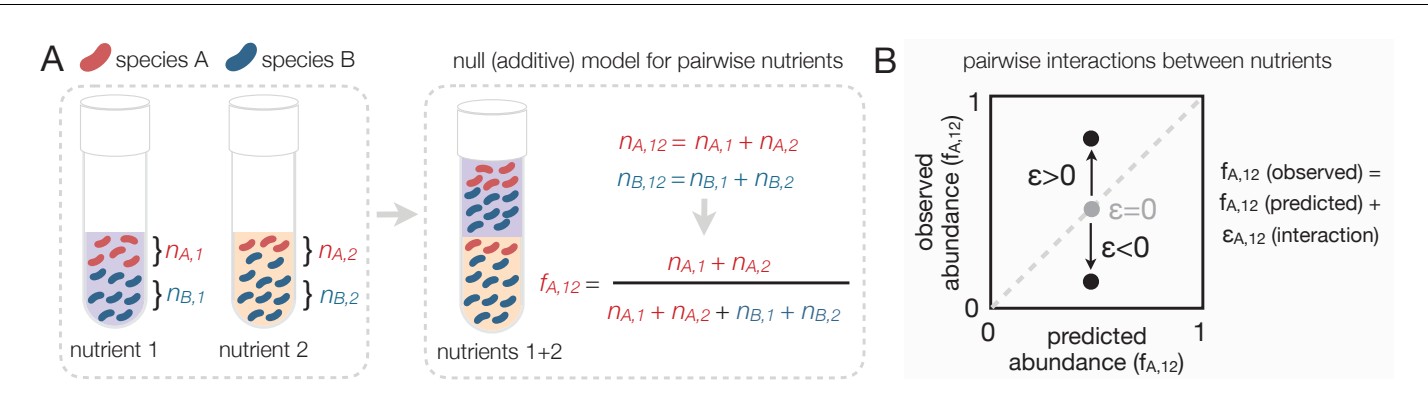

**Figure 1.** Predicting community composition in mixed nutrient environments. (**A**) Community composition in a single nutrient (nutrient 1 or nutrient 2) vs a mixture of nutrients (nutrient 1 + nutrient 2). Assuming that nutrients act independently, the null model predicts that the abundance of each species in the mixture is the sum of its abundance in the single nutrients (i.e. additive). (**B**) Plotting the experimentally measured (observed) relative abundance in the mixed carbon sources against its predicted (from null model) relative abundance reveals the presence or absence of interactions. Any deviation from the identity line (predicted = observed) is the interaction effect ($\varepsilon$). When $\varepsilon = 0$, there is no interaction between nutrients. When $\varepsilon$ is non-zero, community composition is affected by nutrient interactions. If $\varepsilon > 0$, the null model underestimates the abundance. If $\varepsilon < 0$, the model overestimates the abundance.

recruits species just as if the other nutrient were not there. In such case, the abundance of each species in a nutrient mix is the sum of what we would find in the single-nutrient habitats.

Under the null model, the relative abundance of species $i$ in a mix of nutrients 1 and 2 can be written as $f_{i,12}(null) = w_1 f_{i,1} + w_2 f_{i,2}$ where $f_{i,1}$ and $f_{i,2}$ are the relative abundances of $i$ in nutrients 1 and 2, respectively, and $w_1$ and $w_2$ are the relative number of cells in nutrients 1 and 2 (Materials and methods). Any quantitative difference between the null model prediction and the observed composition quantifies an 'interaction' between nutrients. Accounting for the presence of such interactions, the model can be re-written as $f_{i,12} = f_{i,12}(null) + \varepsilon_{i,12}$ where $\varepsilon_{i,12}$ represents the interaction between nutrients 1 and 2 (*Figure 1B*).

## Experimental system

Equipped with this null model, we can now ask to what extent the nutrients recruit species independently in mixed environments. To address this question, we followed a similar enrichment community approach to the one we have used in previous work for studying the self-assembly of replicate microbial communities in a single carbon source (*Goldford et al., 2018*; *Estrela et al., 2020*) (Materials and methods, *Figure 2A*). Briefly, habitats were initially inoculated from two different soil inocula. Communities were then grown in synthetic (M9) minimal media supplemented with either a single carbon source or a mixture of two carbon sources, and serially passaged to fresh medium every 48 hr for a total of 10 transfers (dilution factor = $125\times$ ) (*Figure 2A*). The carbon source pairs consisted of a focal carbon source mixed at equal C-molar concentrations with one of eight additional carbon sources. We previously found that stable multi-species communities routinely assemble in a single carbon source (which is limiting under our conditions), and they converge at the family level in a manner that is largely governed by the carbon source supplied, while the genus or lower level composition is highly variable (*Goldford et al., 2018*). We chose glucose as the focal carbon source because we have previously carried out multiple assembly experiments in this nutrient (*Goldford et al., 2018*; *Estrela et al., 2020*). As the additional carbon sources, we chose nutrients that are simple and metabolically diverse (sugar *vs* acid, that contain a different number of atoms of carbon, and that enter metabolism at different points), namely ribose, fructose, cellobiose, and glycerol (i.e. a pentose, a hexose, a disaccharide, and a sugar alcohol) and fumarate, benzoate, glutamine, and glycine (two organic acids and two amino acids). All carbon sources were also used in single carbon source cultures.

Communities assembled in single sugars contained 5–24 exact sequence variants (ESVs), mainly belonging to the Enterobacteriaceae family (mean relative abundance ± SD of~0.98 ± 0.03), a sugar specialist (*Figure 2—figure supplement 1*). In contrast, communities assembled in organic acids exhibited a higher richness (12–36 ESVs), and unlike in sugars, Enterobacteriaceae were generally rare (mean ± SD ~ 0.06 ± 0.06). Instead, communities were dominated by respiratory bacteria mainly belonging to the Pseudomonadaceae (mean ± SD ~ 0.51 ± 0.25), Moraxellaceae (mean ± SD ~ 0.18 ± 0.21), and Rhizobiaceae (mean ± SD ~ 0.11 ± 0.13) families (*Figure 2—figure supplement 1*). Because of the observed family-level convergence across carbon sources, which is consistent with previous studies (*Goldford et al., 2018*; *Estrela et al., 2020*; *Lu et al., 2018*), we focus our analysis below on family-level abundance.

## The null model of independently acting nutrients explains a high fraction of the variation observed

To investigate the predictive power of the null (additive) model, we compare the predicted and observed relative abundances of each family for each carbon source pair across all experiments. Our results show that the null model predicts reasonably well the family-level abundances on average (Pearson's R = 0.95 and p<0.001; RMSE = 0.073, N = 223) (*Figure 2B*, *Figure 2—figure supplements 2* and *3*). To confirm that the strong predictive power of the null model is not an idiosyncrasy of using glucose as the focal carbon source in the pairs, we repeated the same experiment with succinate (an organic acid) as the focal carbon source. Although the correlation between observed and predicted abundance is lower than for glucose, the null additive model is still predictive (Pearson's R = 0.87 and p<0.001; RMSE = 0.094; N = 257) (*Figure 2B*).

This result seems to indicate that, at the family level, a simple model that assumes that nutrients act independently can predict community composition in a pair of nutrients (for an analysis of this

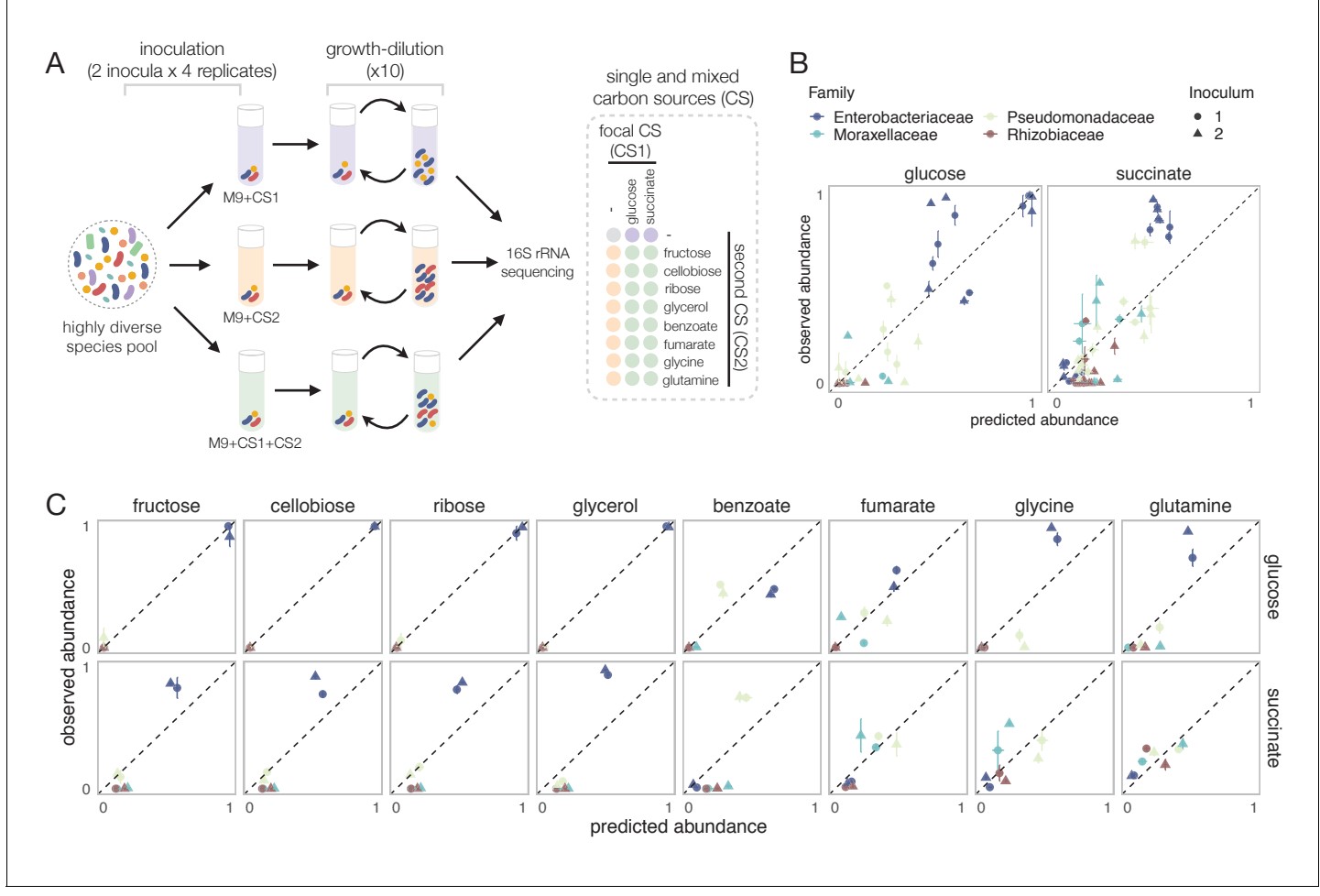

**Figure 2.** Systematic deviations from the null prediction reveals that some nutrients interact to shape community assembly. (**A**) Schematic of experimental design. Two different soil samples were inoculated in minimal M9 medium supplemented with either a single carbon source (CS1 or CS2) or a mixture of two carbon sources (CS1 + CS2) (three to four replicates each). Communities were propagated into fresh media every 48 hr for 10 transfers and then sequenced to assess community composition. Carbon source mixtures consisted of a focal carbon source (CS1; glucose or succinate) mixed with a second carbon source (CS2). (**B, C**) For each pair of carbon sources, we show the experimentally observed and predicted (by the null additive model) relative abundance of each family in the mixture. Any deviation from the identity line (predicted = observed) reveals an interaction effect. Only the four most abundant families are shown. Error bars represent mean ± SE.

The online version of this article includes the following figure supplement(s) for figure 2:

**Figure supplement 1.** Community assembly in a single carbon source.

**Figure supplement 2.** Community assembly in a mixture of two carbon sources.

**Figure supplement 3.** Systematic deviations from the null (additive) prediction reveal interactions between nutrients.

**Figure supplement 4.** Comparison of the observed relative abundance and abundance predicted by the null model.

**Figure supplement 5.** Community yield in each single carbon source.

point at the genus and ESV level, see *Figure 2—figure supplement 4*). However, when we looked at this more closely and broke down our results by carbon source and family, we found consistent and systematic deviations from the null model (*Figure 2C*). For example, across all succinate–sugar pairs, Enterobacteriaceae are significantly more abundant than predicted by the null model (one-tailed paired t-test, N = 8, p<0.05 based on 1000 permutations; see Materials and methods), while both Rhizobiaceae and Moraxellaceae are less abundant than predicted (one-tailed paired t-test, N = 8, p<0.05 based on 1000 permutations; see Materials and methods) (*Figure 2C*). The null 'inter-action-free' model also predicts species abundance better in certain carbon source combinations (e.g. glucose + ribose) than in others (e.g. glucose + glutamine) (*Figure 2C*). The existence of

systematic deviations from the null prediction reveals that some nutrient pairs do not recruit families independently, but instead 'interact' with each other to affect the abundance of specific families.

## A simple dominance rule in mixed nutrient environments: sugars generally dominate organic acids

To map the regularities in nutrient interactions observed, we next sought to characterize the nature of these interactions for each carbon source pair and every family. One helpful way of visualizing nutrient interactions is to draw the pairwise abundance landscape for each species and carbon source pair (*Figure 3A*). For instance, a species could be either more abundant in a pair of nutrients than it is in any of them independently (synergy). Or it could be less abundant than it is in any of the two (antagonism). Dominance is a less extreme interaction that can be visualized by the pushing of a species abundance toward the value observed in one of the two nutrients and away from the average, thus overriding the effect of the second available nutrient (*Figure 3A*).

When the interaction is positive ($\varepsilon > 0$), the dominant nutrient is the one where the family grew to a higher abundance. When the interaction is negative ($\varepsilon < 0$), the dominant nutrient is the one where the species grew less well. Mathematically, dominance occurs when $|\varepsilon| > 0$ and $min(f_{i,1}, f_{i,2}) \leq f_{i,12} \leq max(f_{i,1}, f_{i,2})$, while synergy and antagonism (forms of super-dominance) occur when $|\varepsilon| > 0$ and $f_{i,12} > max(f_{i,1}, f_{i,2})$ and $f_{i,12} < min(f_{i,1}, f_{i,2})$, respectively (Materials and methods). *Figure 3B* shows representative examples of dominant carbon source interactions. For instance, Moraxellaceae and Rhizobiaceae grow strongly on succinate, but they are not found in fructose. When fructose is mixed with succinate, both families drop dramatically in abundance, despite their high fitness in succinate alone. Interestingly, however, the dominance of fructose over succinate is not observed for all families: those two nutrients do not interact on Pseudomonadaceae, whose abundance is well predicted by the null model. Using this framework, we then systematically quantified the prevalence of dominance, antagonism, and synergy between nutrients for each family (*Figure 3—figure supplement 1A*). While 66% of the nutrient pair combinations exhibited no significant interaction, dominance was by far the most common interaction amongst those that interacted (75%, *Figure 3—figure supplement 1A*). It occurred predominantly in the sugar–acid pairs, and to a lesser extent in the acid–acid pairs, and only rarely in the sugar–sugar pairs (*Figure 3—figure supplement 1B*). This result strongly suggests that nutrient interactions are not random but do have a specific structure that is conserved at the family level (*Figure 3—figure supplement 1C*).

To systematically characterize and quantify nutrient dominance, we developed a dominance index ($\delta$) (Materials and methods). For visualization purposes, the dominance index for the sugar–acid pairs (we will discuss the acid–acid pairs later) is written as $\delta_i = -|\varepsilon_{12}|$ when the sugar dominates and as $\delta_i = |\varepsilon_{12}|$ when the acid dominates. If $\varepsilon_{12} = 0$, then $\delta_i = 0$. That is, in the absence of interaction between nutrients, there is no dominance. By plotting the dominance index for each pair of nutrients and each family, we observe a generic pattern of dominance of sugars over acids (*Figure 3C*). The families Moraxellaceae or Rhizobiaceae are recruited to the community by most organic acids in isolation, but they are not found in most sugar communities. When sugars and organic acids are mixed together, the sugar dominates and both families are at much lower abundances (by ~6-fold in the case of Moraxellaceae and ~114-fold in Rhizobiaceae) than expected by the null model, even though the organic acid where they thrived is present in the environment. Consistent with this pattern, we found that pairs of more-similar nutrients (a pair of sugars or a pair of organic acids) were significantly better predicted by the null model than mixed organic acid–sugar pairs (*Figure 3D*). No generic pattern of dominance was observed in the acid–acid mixtures (*Figure 3—figure supplement 2*). When we examine interactions and dominance at the genus level, we find that sugars do not exhibit the same dominance for all genera within the same family (*Figure 3—figure supplements 3–4*). This result is consistent with the convergence of community structure at the family level (despite substantial variation at lower levels of taxonomy), which we have reported for communities assembled in a single nutrient (*Goldford et al., 2018*; *Estrela et al., 2020*). Together, these results indicate that interactions between nutrients are not universal, but rather they are conserved at the family-level.

## An extension of the null consumer-resource model with an asymmetry in nutrient benefits recapitulates the dominance pattern observed

Our findings pose intriguing questions about the mechanisms behind the nutrient interaction patterns we have observed. For instance, is it reasonable to expect that the additive null model should have worked as well as it did, and better at the family than at the species level? Why are pairs of more-similar nutrients better explained by the null model than pairs of more dissimilar nutrients? What may explain why nutrients dominate over others at the family level? And why do sugars generally dominate organic acids for most families?

We have previously shown that many of the properties of our experimental enrichment communities reflect the generic emergent behavior of consumer-resource models (*Goldford et al., 2018*; *Marsland et al., 2019*), and subsequent work extended this finding to complex natural communities (*Marsland et al., 2020a*). We thus sought to ask whether our observations regarding the assembly of communities in pairs of resources are similarly reflecting a generic emergent property of consumer-resource models. To address this question, we followed the same procedure as we and others have done in previous work (*Goldford et al., 2018*; *Marsland et al., 2020a*; *Marsland et al., 2020b*; *Serván and Allesina, 2020*), and simulated the top-down assembly of microbial communities in pairs vs single nutrients using a recently developed Microbial Consumer Resource Model (MiCRM) (*Goldford et al., 2018*; *Marsland et al., 2019*; *Marsland et al., 2020a*) (see Materials and methods). The MiCRM differs from the classical MacArthur-Levins model (*MacArthur, 1970*) in that it includes metabolic cross-feeding in a manner that preserves thermodynamic balance. The model and the details of the simulations are described in the Materials and methods section. In brief, 200 species are seeded into each habitat at the start of a simulation. Each of these is represented by a different vector of resource uptake rates. These vectors are randomly sampled in a manner that captures the existence of two functional guilds, each of which specializes in a different group of resources (e.g. sugars *vs* organic acids) (*Figure 4A*). Members of the family specializing on sugars (i.e. the Enterobacteriaceae) have on average a higher uptake rate on each sugar whereas members of the family specializing on acids (i.e. the Pseudomonadaceae) have on average a higher uptake rate on each acid. The magnitude of specialization by each family on its preferred resource type is tuned by two parameters, $q_A$ and $q_S$, which modulate the mean and variance of the probability distribution from which the uptake rates are sampled (see Materials and methods for more details). We note that this specialization is quantitative rather than discrete, as all species are assumed to be able to consume all of the resources (a point that is in general consistent with our experimental findings [*Figure 4—figure supplement 1*]). Communities are allowed to find a dynamical equilibrium, at which point we stop the simulation. In total, and in order to get to generic behavior, we generated 100 simulations each with a different random set of species (Materials and methods).

A generic property of these simulations is that the species-level community composition on mixtures of two limiting nutrients is reasonably well described by the additive null model (Pearson's R = 0.7 and p<0.001; RMSE = 0.097; N = 2440) (*Figure 4B*; Materials and methods), which is consistent with previous consumer-resource modelling work (*Marsland et al., 2020a*). In addition, when we group species by the functional groups they belong to (i.e. family), the predictive ability of the additive null model improves (Pearson's R = 0.99 and p<0.001; RMSE = 0.03; N = 414) (*Figure 4B*), a point that is consistent with our experimental findings (*Figure 2—figure supplement 4*). This family-level additivity holds when communities are randomly colonized by a different set of species (*Figure 4—figure supplement 2*), suggesting that family-level additivity is robust to species-level taxonomic variability. The predictive accuracy of our null model is, however, influenced by the level of resource specialization. The less specialized (i.e. more generalist) the families are, the lower the predictive power of the null additive model (*Figure 4—figure supplement 3*).

By contrast, the simulated communities do not exhibit any systematic dominance, neither at the species nor at the family level (*Figure 4B*). What feature of the MiCRM might be causing us to miss this experimental behavior? One assumption of the model, which we had made for the sake of simplicity and for consistency with previous work, is that all nutrients are equally valuable for the microbes that specialize on them. In other words, the benefits of growing in each type of nutrient are symmetric. Yet, this assumption is not really consistent with the empirical reality that glucose specialists, such as Enterobacteriaceae, grow more strongly in sugars than organic acid specialists

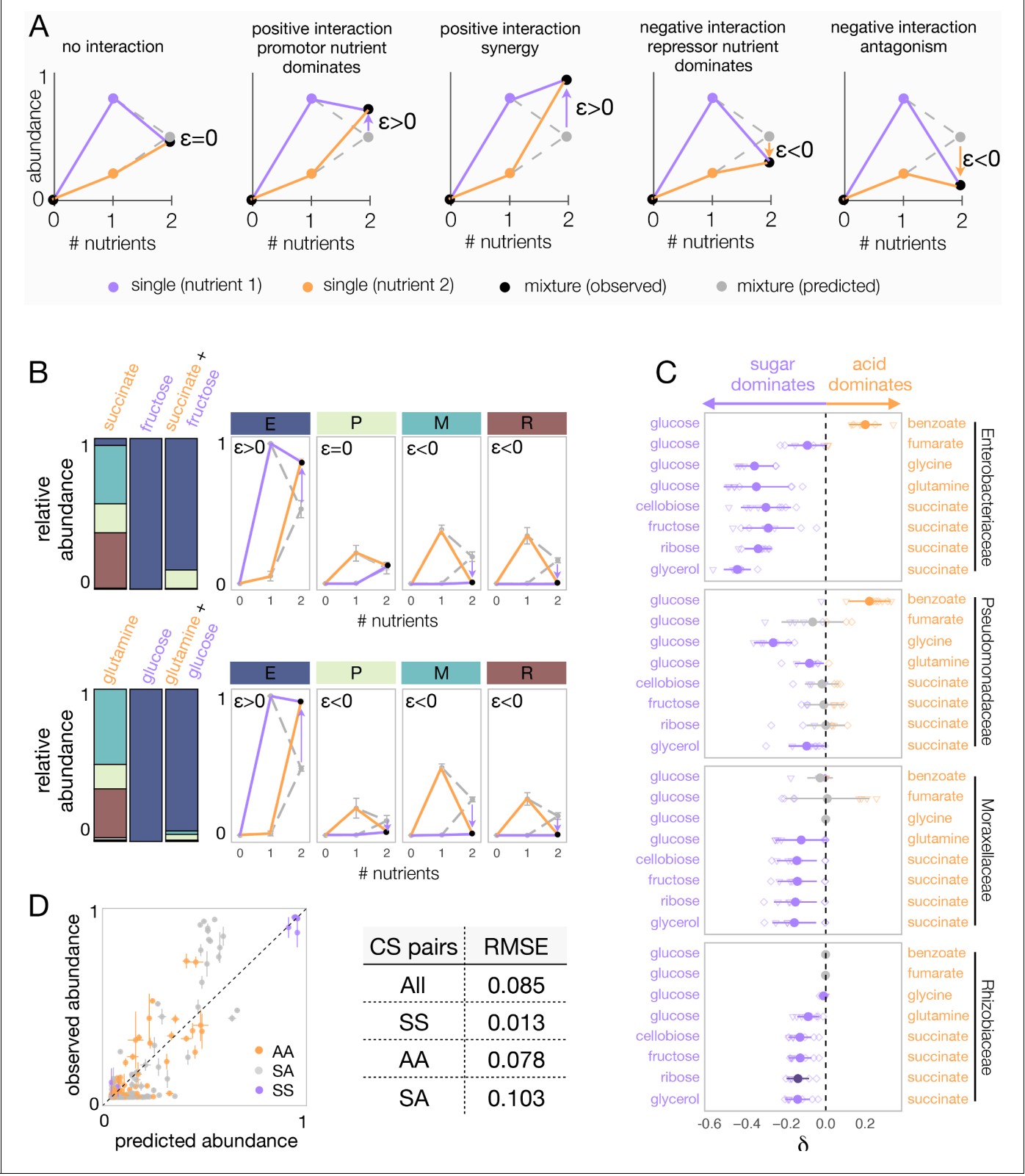

**Figure 3.** Sugars generally dominate over organic acids. (A) Detecting interactions and hierarchies of dominance between nutrients on microbial community composition. Drawing the single and pairwise abundance landscapes for each species allows us to visualise interactions between nutrients. Multiple types of interactions are possible, including dominance, synergy, and antagonism. Interactions occur when ε is significantly different from 0 (Materials and methods). Synergy (antagonism) occurs when the abundance in the mixture is greater (lower) than the abundances in any of the single

*Figure 3 continued on next page*

*Figure 3 continued*

nutrients independently (Materials and methods). Dominance occurs when the abundance in the mixture is closer or similar to the abundance in one of the singles. The landscape also allows us to identify which carbon source has a dominating effect within the pair. When ε > 0, the growth-promoting nutrient dominates and has an overriding effect in the community composition. In contrast, when ε < 0, the growth-repressing nutrient dominates. (B) Two examples of nutrient interactions (succinate + fructose and glucose + glutamine) exhibiting sugar dominance. Barplots show a representative replicate from one of the inocula (*Figure 2—figure supplements 1–2*). For instance, the landscape for succinate-fructose shows that fructose overrides the effect of succinate by promoting Enterobacteriaceae (E), and repressing Moraxellaceae (M) and Rhizobiaceae (R) (purple arrows), whereas no interaction effect is observed for Pseudomonadaceae (P). Error bars represent mean ± SD of the four replicates. (C) Dominance index for the eight sugar–acid pairs and the four dominant families. Filled circles show the mean ± SD of the two inocula × four replicates for each pair of nutrients, and open symbols show all eight independent replicates (different shapes for different inocula), except for glycine pairs where N = 6. Purple indicates that the sugar dominates while orange indicates that the acid dominates. Lighter purple and orange indicate dominance, while darker purple and orange indicate super-dominance (synergy or antagonism). An interaction occurs when the abundance is greater (ε > 0) or lower (ε < 0) in the carbon source mixture than predicted by the null model (one-tailed paired t-test, p<0.05, N = 8, based on 1000 permutations; Materials and methods). In gray are shown cases where there is no interaction, or when dominance is undefined because the two inocula exhibit opposite dominant nutrient (in which case δ is shown as both –δ and +δ). (D) Predicted vs observed family-level abundance. For each pair of carbon sources (CS), shown is the experimentally observed and predicted (by the null model) relative abundance of each family in the mixed carbon sources. Any deviation from the identity line (predicted = observed) is the interaction effect. The colors show whether the carbon source pairs are sugar–sugar (SS), acid–acid (AA), or sugar–acid (SA). Error bars represent mean ± SE. Table shows RMSE for each carbon source pair type.

The online version of this article includes the following figure supplement(s) for figure 3:

**Figure supplement 1.** Dominance is the most common type of nutrient interaction, especially in the sugar–acid mixtures.
**Figure supplement 2.** Family-level dominance for mixtures of acid–acid and sugar–sugar.
**Figure supplement 3.** Patterns of nutrient interaction at the genus level.
**Figure supplement 4.** The systematic dominance of sugars observed at the family level does not apply to the genus level.

do on organic acids. This is illustrated in *Figure 4C*, where we plot the growth advantage for seven Enterobacteriaceae isolates in sugar media *vs* the growth advantage of Pseudomonadaceae, Rhizobiaceae, and Moraxellaceae isolates in organic acids.

We postulated that including this asymmetry may unbalance the competition for resources and give rise to nutrient dominance at the family level, as the family that lies on the winning side of that asymmetry may leverage its enhanced competitive ability in the most valuable nutrient to displace the losing family from its lower-value nutrient niche. To test this intuition, we relaxed the symmetry in resource value that was imposed by default in the model, and repeated our simulations for different levels of nutrient value asymmetry (the simulations still include facilitation via metabolite secretion, as we had done in all prior simulations) (*Figure 4A*, *Figure 4—figure supplement 4*). As we show in *Figure 4D*, and consistent with our intuition, nutrient dominance at the family level may emerge as a generic property of microbial consumer-resource models when a nutrient is substantially more valuable than the other. Reassuringly, our experiments indicate that dominance is generally favorable to the taxa that benefits from growth asymmetry for example to Enterobacteriaceae in sugar–acid mixes, and unfavorable to families in the losing end of growth asymmetry (Pseudomonadaceae, Rhizobiaceae, and Moraxellaceae) in those same environments. This observation is consistent with the behavior of the model (*Figure 4D*).

Our consumer-resource model shows that dominance is a general outcome of consumer-resource interactions when there exists an asymmetry in nutrient benefit (an asymmetry that is indeed observed for the families in our communities), but other mechanisms of dominance may be at play too. For instance, one plausible mechanism that could lead to dominance is oxygen limitation, in particular if different carbon sources were to have different oxygen requirements (*Hempfling and Mainzer, 1975*; *Skrinde and Bhagat, 1982*). To explore this idea of asymmetric oxygen demands, we used flux-balance analysis (FBA) to determine the oxygen demands of growth on each of the single-carbon sources. We found that, except for benzoate, all carbon sources have similar oxygen demands (*Figure 4—figure supplement 5*). This does not rule out, however, the possibility that kinetics of growth and oxygen uptake may still contribute to oxygen depletion in a manner that may further stimulate dominance (in addition to the asymmetric resource benefits we report in *Figure 4*).

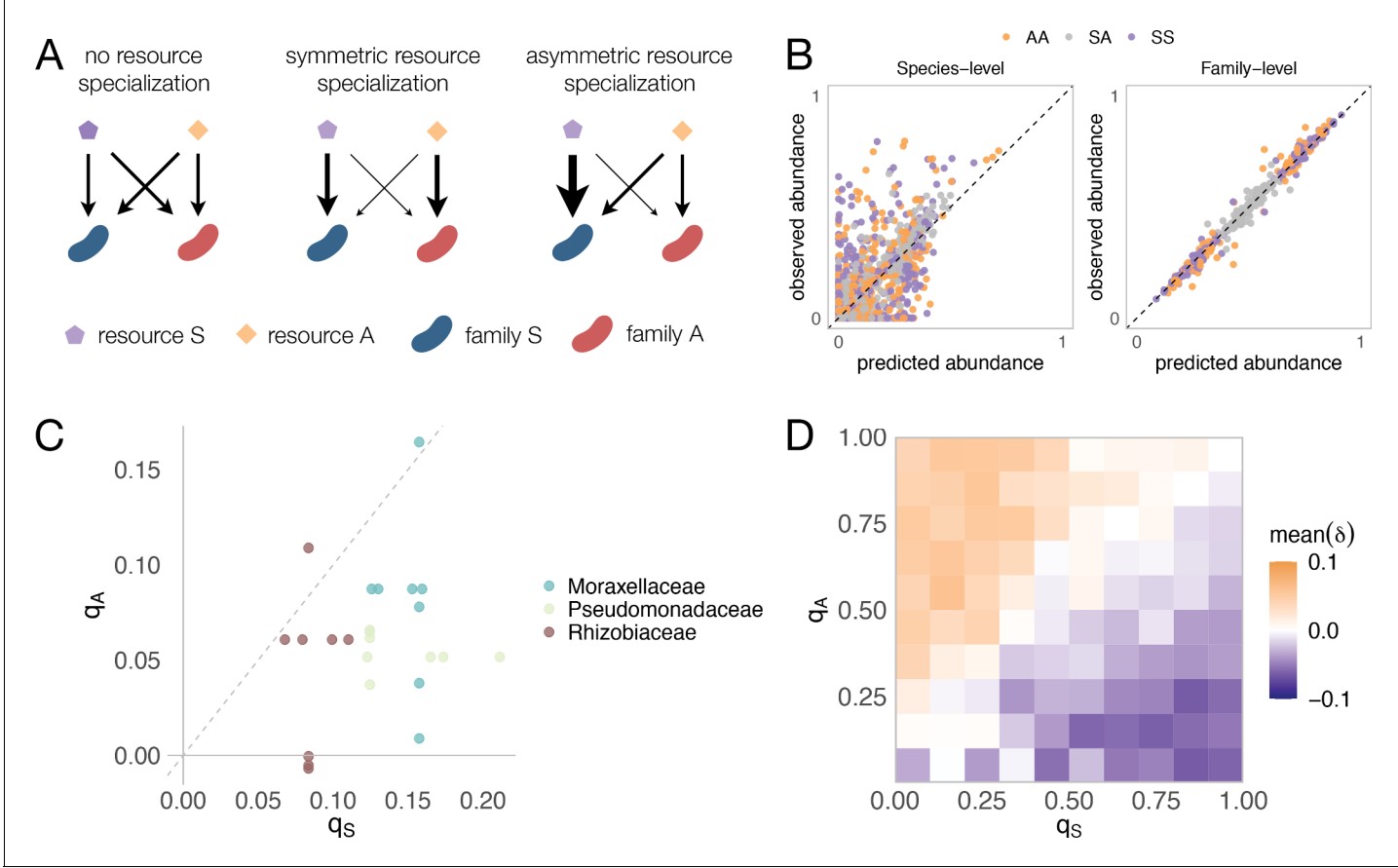

**Figure 4.** Family-level asymmetry in nutrient benefits can lead to dominance. (A) Schematic illustrating different scenarios of nutrient preference. There are two families ($F_S$ and $F_A$) and two resource classes ($R_S$ and $R_A$). Without resource specialization, $F_S$ and $F_A$ have equal access to $R_S$ and $R_A$. With symmetric specialization, each family prefers its own resource class with the same strength. With asymmetric specialization, one family ($F_S$) has better access to its own resource class ($R_S$) relative to that of the other family ($F_A$) on its own resource class ($R_A$). (B) A mechanistically explicit consumer-resource model that incorporates resource competition, resource specialization and nonspecific cross-feeding (Materials and methods) recovers the predicted additivity pattern at both the species (left) and family (right) level of taxonomic organization. The observed relative abundance of each species or family in 300 communities grown on a different pair of nutrients (100 AA, 100 SS, and 100 SA) is plotted against the abundance predicted from the same communities grown on each of the relevant single nutrients (S, A). Each family specializes equally on its preferred nutrient ($q_S = q_A = 0.9$) as in previous work (*Marsland et al., 2020a*). In *Figure 4—figure supplement 4*, we illustrate representative consumption matrices for different choices of $q_A$ and $q_S$. (C) 22 strains were isolated from the assembled communities and their growth rates on minimal M9 media supplemented with one the 10 carbon sources were measured. $q_S$ represents the growth rate advantage of Enterobacteriaceae on sugars relative to the other dominant family (colored), while $q_A$ represents the growth rate advantage of the other family on the acids relative to Enterobacteriaceae (Materials and methods). When $q_S$ is positive, Enterobacteriaceae grow faster on the sugar than the other family, while when $q_S$ is negative, Enterobacteriaceae grow more slowly on the sugar than the other family. When $q_A$ is positive, the other family grows faster on the acid than Enterobacteriaceae, while when $q_A$ is negative, the other family grows more slowly on the acid than Enterobacteriaceae. Each dot corresponds to a sugar-acid pair for a Enterobacteriaceae-other family pair (n = 24). The growth rate advantage of Enterobacteriaceae on sugars is significantly greater than the growth rate advantage of the other families on acids (i.e. $q_S > q_A$, mean of differences = 0.069, paired t-test, n = 24, p-value<0.0001). (D) Here we repeat the same simulation as shown in (B), this time using different combinations of $q_A$ and $q_S$ (0.05, 0.15, 0.25, 0.35, 0.45, 0.55, 0.65, 0.75, 0.85, 0.95). Heatmap shows the mean dominance level ($\delta$) for different combinations of $q_A$ and $q_S$. When $\delta < 0$, the sugar dominates (purple); when $\delta > 0$, the acid dominates (orange). The online version of this article includes the following figure supplement(s) for figure 4:

**Figure supplement 1.** Enterobacteriaceae generally have a strong growth advantage in sugars.
**Figure supplement 2.** Stochastic colonization has no qualitative effect on the pattern of additivity found using a Microbial Consumer Resource Model.
**Figure supplement 3.** The predictive accuracy of the null model decreases with lower levels of resource specialization.
**Figure supplement 4.** Consumption matrices for different patterns of nutrient preference between families used in the consumer-resource model simulations.
**Figure supplement 5.** Oxygen demands are similar across the different carbon sources.

## Discussion

Our analysis indicates that our empirical observations regarding the assembly of microbial communities in nutrient mixes are consistent with generic behavior of consumer-resource models. Based on this finding, we cautiously suggest that family-level asymmetries in nutrient uptake rates may be a possible mechanism for the general nutrient dominance patterns we have observed, and that a null additive model is in general a good first approximation for the assembly of microbial communities in simple nutrient mixtures (a pattern that is consistent with previous work [*Faith et al., 2011*; *Enke et al., 2019*]). It is important to recognize, however, that other explanations and mechanisms of dominance may be at play too. Generally, these can be split into two main categories: asymmetries in how species respond to the provided nutrient and asymmetries that emerge as a result of the constructed environment. Below, we discuss several specific mechanisms that may contribute to each of these.

Our null model (and consumer-resource simulations) assumes, by definition, that the growth of a single species on a mixture of nutrients (in terms of growth rate and yield) will be the aggregate sum of the growth on each nutrient alone. Multiple mechanisms, however, could lead to violations of this assumption. Firstly, a species might not consume both nutrients simultaneously but may instead consume them sequentially, or diauxically, resulting in fluctuations in the effective resource specialization of each species (*Monod, 1942*; *Lendenmann et al., 2000*; *Erickson et al., 2017*; *Pacciani-Mori et al., 2020*). Secondly, even if a species is co-utilizing both nutrients, the biomass yields may not be additive, due to synergistic effects of using different nutrients for different cellular functions (such as energy versus biomass or for synthesis of different biomass precursors) (*Lendenmann et al., 1996*; *Pacheco et al., 2019*; *Wang et al., 2019*). Thirdly, a molecule that can be used as a nutrient by one species may have an inhibitory effect on another species, for example benzoate is known to have antimicrobial activities against some bacteria (which may explain why benzoate dominates sugars for some families in *Figure 3C*; *Stanojevic et al., 2009*). The growth dynamics on mixtures of carbon sources have been extensively characterized in simple sugars for a few model organisms (such as *Escherichia coli*, *Bacillus subtilis*, and *Pseudomonas aeruginosa*), but we still lack a systematic understanding of mixed-substrate growth across taxa and environment (*Harder and Dijkhuizen, 1982*; *Görke and Stülke, 2008*; *Bajic and Sanchez, 2020*). Systematically mapping mixed-resource utilization strategies represents an exciting direction for future work and would allow us to better predict the effects of environmental complexity on the emergent properties of complex microbial communities.

Importantly, even if species respond to the supplied pair of nutrients in an additive manner, niche construction (and thus the interactions between species) may not be additive. For example, species may secrete secondary metabolites or antimicrobial agents on nutrient mixtures, which may interact with each other (*Sánchez et al., 2010*; *Mendonca et al., 2020*; *Fujiwara et al., 2020*). Moreover, cellular growth can change other physico-chemical properties of the environment aside from carbon source availability, such as by changing the pH, the accessibility of non-carbon source nutrients leading to co-limitation, or oxygen availability (*Harpole et al., 2011*; *Cremer et al., 2017*; *Sánchez-Clemente et al., 2020*).

The wealth of independent mechanisms that may contribute to nutrient dominance illustrates the potential importance of this phenomenon. Quantitatively elucidating the specific mechanisms that may explain the individual patterns of nutrient interactions (or lack thereof) for each family and in each pair of nutrients would require us to measure the amounts of all nutrients secreted by every species in each environment over time (i.e. in each nutrient and in each pair) and then characterize the growth curves of all species in those nutrients. Although such monumental effort is beyond the scope of this paper, we hope that our findings and methodology will be a stepping stone towards elucidating how microbial communities assemble in complex nutrient mixtures and that they will stimulate further theoretical and empirical work. We propose that top-down community assembly in combinatorially reconstructed nutrient environments can be a helpful approach not only to understand the origins of microbial biodiversity, but also to learn how to manipulate existing microbiomes by rationally modulating nutrient availability.

# Materials and methods

**Key resources table**

| Reagent type (species) or resource | Designation | Source or reference | Identifiers | Additional information |
|---|---|---|---|---|
| Chemical compound, drug | D-Glucose | VWR | 0188–500 | |
| Chemical compound, drug | D-Cellobiose | Sigma | 22150–10G | |
| Chemical compound, drug | D-Fructose | Acros Organics | 161355000 | |
| Chemical compound, drug | D-Ribose | Acros Organics | AC132361000 | |
| Chemical compound, drug | Glycerol (80%, w/v) | Teknova | G8797 | |
| Chemical compound, drug | Sodium Succinate hexahydrate | Alfa Aesar | 419A3 | |
| Chemical compound, drug | Sodium hydrogen fumarate | Alfa Aesar | B24683 | |
| Chemical compound, drug | Sodium benzoate | Alfa Aesar | A15946 | |
| Chemical compound, drug | L-Glutamine 200 mM (29.23 mg/mL) | Sigma | G7513-100ML | |
| Chemical compound, drug | Glycine | Sigma | G7126-100G | |
| Software, algorithm | R | R Development Core Team, 2017. R: A language and environment for statistical computing. R Foundation for Statistical Computing, Vienna, Austria. https://www.r-project.org/ | RRID:SCR_001905 | R version 3.4.3 |
| Software, algorithm | DADA2 | Callahan BJ, McMurdie PJ, Rosen MJ, Han AW, Johnson AJA, Holmes SP. DADA2: High-resolution sample inference from Illumina amplicon data. Nat Methods. 2016;13: 581–583. | | Version 1.6.0 |
| Software, algorithm | Community simulator | Marsland R, Cui W, Goldford J, Mehta P. The Community Simulator: A Python package for microbial ecology. PLoS One. 2020;15: e0230430. | | Version 1.0 |

*Continued on next page*

*Continued*

| Reagent type (species) or resource | Designation | Source or reference | Identifiers | Additional information |
|---|---|---|---|---|
| Software, algorithm | COBRApy | Ebrahim A, Lerman JA, Palsson BO, Hyduke DR. COBRApy: COnstraints-Based Reconstruction and Analysis for Python. BMC Syst Biol. 2013;7: 74. | RRID:SCR_012096 | Version 0.17.0 |

## Null model for relative abundance

Let us consider a simple scenario of two co-cultures of species A and B growing together in two separate demes, each containing a single nutrient (labeled 1 and 2). The fractions of A and B in nutrient/deme one are $f_{A,1} = n_{A,1}/(n_{A,1} + n_{B,1})$ and $f_{B,1} = n_{B,1}/(n_{A,1} + n_{B,1})$, respectively, and similarly, the fractions of A and B in nutrient/deme two are $f_{A,2} = n_{A,2}/(n_{A,2} + n_{B,2})$ and $f_{B,2} = n_{B,2}/(n_{A,2} + n_{B,2})$ (where $n$ is the total number of cells of species A or B). If we consider the two-deme system as a whole (i.e. if we pool together the amount of species in each nutrient/deme), the fractions of A and B in the mixture are given by: $f_{A,12} = (n_{A,1} + n_{A,2})/(n_{A,1} + n_{B,1} + n_{A,2} + n_{B,2})$ and $f_{B,12} = (n_{B,1} + n_{B,2})/(n_{A,1} + n_{B,1} + n_{A,2} + n_{B,2})$.

We can define $n_{t,1} = n_{A,1} + n_{B,1}$ and $n_{t,2} = n_{A,2} + n_{B,2}$ as the total number of cells in the nutrient demes 1 and 2, respectively. We can thus write $f_{A,12} = (n_{A,1} + n_{A,2})/(n_{t,1} + n_{t,2})$. Defining $w_1 = n_{t,1}/(n_{t,1} + n_{t,2})$ and $w_2 = n_{t,2}/(n_{t,1} + n_{t,2})$, it is straightforward to show that: $f_{A,12} = w_1 f_{A,1} + w_2 f_{A,2}$. By the same reasoning, we find that $f_{B,12} = w_1 f_{B,1} + w_2 f_{B,2}$.

## Sample collection

Soil samples were collected from two different natural sites in West Haven (CT, USA), with sterilized equipment, and placed into sterile bottles. Once in the lab, 5 g of each soil sample were then transferred to 250 mL flasks and soaked into 50 mL of sterile 1× phosphate buffer saline supplemented with 200 µg/mL cycloheximide (Sigma, C7698) to inhibit eukaryotic growth. The soil suspension was well mixed and allowed to sit for 48 hr at room temperature. After 48 hr, samples of the supernatant solution containing the 'source' soil microbiome were used as inocula for the experiment or stored at −80°C with 40% glycerol.

## Preparation of media plates

Carbon source (CS) stock solutions (*Supplementary file 1a*) were prepared at 0.7 C-mol/L (10×) and sterilized through 0.22 µm filters (Millipore). Carbon sources were aliquoted into 96 deep-well plates (VWR) as single CS or mixed in pairs at 1:1 (vol:vol) and stored at −20°C. The carbon sources were adjusted to equal C-molar concentrations because carbon is the main limiting factor. To keep the total amount of carbon constant across all treatments, pairs contained half the amount of each carbon source compared to their respective single CS. Synthetic minimal growth media was prepared from concentrated stocks of M9 salts, MgSO$_4$, CaCl$_2$, and 0.07 C-mol/L (final concentration) of single or pairs of CS. The final pH of all growth media is shown in *Supplementary file 1a*.

## Community assembly experiment

Starting inocula were obtained directly from the 'source' soil microbiome solution by inoculating 40 µL into 500 µL culture media prepared as indicated above. For each sample and carbon source, 4 µL of the culture medium was dispensed into fresh media plates containing the different single or pairs of CS in quadruplicate. Bacterial cultures were allowed to grow for 48 hr at 30°C in static broth in 96 deep-well plates (VWR). After 48 hr, each culture was homogenized by pipetting up and down 10 times before transferring 4 µL into 500 µL of fresh media, and cells were allowed to grow again.

Cultures were passaged 10 times (~70 generations). OD620 was measured after 48 hr growth. Samples were frozen at −80°C with 40% glycerol.

## DNA extraction, library preparation, and sequencing

Samples were centrifuged for 40 min at 3500 rpm, and the pellet was stored at −80°C until DNA extraction. DNA extraction was performed with the DNeasy 96 Blood and Tissue kit for animal tissues (QIAGEN), as described in the kit protocol, including the pre-treatment step for Gram-positive bacteria. DNA concentration was quantified using the Quan-iTPicoGreen dsDNA Assay kit (Molecular Probes, Inc), and the samples were normalized to 5 ng/μL before sequencing. The 16S rRNA gene amplicon library preparation and sequencing were performed by Microbiome Insights, Vancouver, Canada (https://microbiomeinsights.com/). For the library preparation, PCR was done with dual-barcoded primers (*Kozich et al., 2013*), targeting the 16S V4 region, and the PCR were cleaned up and normalized using the high-throughput SequalPrep 96-well Plate Kit. Samples were sequenced on the Illumina MiSeq using the 300 bp paired-end kit v3.chemistry.

## Taxonomy assignment

The taxonomy assignment was performed as described in previous work (*Estrela et al., 2020*). Following sequencing, the raw sequencing reads were processed, including demultiplexing and removing the barcodes, indexes, and primers, using QIIME (version1.9, [*Caporaso et al., 2010*]), generating fastq files with the forward and reverse reads. DADA2 (version 1.6.0) was then used to infer ESVs (*Callahan et al., 2016*) . Briefly, the forward and reverse reads were trimmed at position 240 and 160, respectively, and then merged with a minimum overlap of 100 bp. All other parameters were set to the DADA2 default values. Chimeras were removed using the 'consensus' method in DADA2. The taxonomy of each ESV was then assigned using the naïve Bayesian classifier method (*Wang et al., 2007*) and the Silva reference database version (*Quast et al., 2013*) as described in DADA2. The analysis was performed on samples rarefied to 10,779 reads.

## Quantification of total abundances, interactions, and dominance

We used OD620 after the 48 hr growth cycle as a proxy for total population size (community biomass) (*Figure 2—figure supplement 5*). The predicted relative abundance of species $i$ in a mix of nutrients 1 and 2 was then calculated as $f_{i,12}$(null) = $w_1 f_{i,1} + w_2 f_{i,2}$ where $f_{i,1}$ and $f_{i,2}$ are the relative abundances of $i$ in nutrients 1 and 2, respectively, and $w_1$ = (OD620$_1$/(OD620$_1$ +OD620$_2$)) and $w_2$ = (OD620$_2$/(OD620$_1$ +OD620$_2$)). In *Figures 2* and *3D*, *Figure 2—figure supplements 3–4*, $f_{i,12}$(null) is calculated as the mean of the two single carbon source-replicate pairwise combinations (N = 16). Pearson's R was calculated using the R function 'cor.test' from the 'stats' package, and the RMSE was calculated using the 'rmse' function from the 'Metrics' package.

To determine whether an interaction between nutrients exists (i.e. $\varepsilon \neq f_{i,12}$ - $f_{i,12}$(null)), we assess whether the abundance observed in the carbon source mixture is significantly greater or lower than the abundance predicted by the null additive model (i.e. $\varepsilon > 0$ or $\varepsilon < 0$, respectively) (one-tailed paired t-test). More specifically, considering the two inocula and four replicates per carbon source, the family-level analysis was done as follow. For each carbon source pair and inoculum, four predicted pairs are formed by randomly pairing one replicate of each carbon source. Four unique observed vs predicted pairs are then randomly formed from the 64 possible combinations (i.e. from the four single nutrient 1 × four single nutrient 2 × four mixed nutrient 12). Up to this point, the pairs are formed for each inoculum separately, in other words, there is no cross-inocula pairing. Once all N=8 pairs are formed (i.e. N = 4 pairs per inoculum), they are pooled to perform the one-tailed paired t-test. The N = 8 pairs are then randomly permuted 1000 times, determining the t-statistic for each permutation. We establish a 95% confidence threshold for the t-statistic. The effect observed is statistically significant (i.e. an interaction exists) if a significant difference is found in more than 95% of the permuted pairs. At the genus level, the analysis was performed in a similar way, except that the two inocula were kept separately. This is because, compared to families, the likelihood that genera that are sampled in one of the inocula is sampled in the other inoculum is much lower.

Once an interaction has been identified (i.e. $|\varepsilon| > 0$), we can determine the type of interaction formed (*Figure 3A*). Synergy and antagonism (which are forms of super-dominance) occur when

$f_{i,12} > max(f_{i,1}, f_{i,2})$ and $f_{i,12} < min(f_{i,1}, f_{i,2})$, respectively, while dominance occurs when $min(f_{i,1}, f_{i,2}) <= f_{i,12} <= max(f_{i,1}, f_{i,2})$ (Welch two sample t-test, p<0.05). When $\varepsilon > 0$, the nutrient with greater abundance dominates; when $\varepsilon < 0$, the nutrient with lower abundance dominates. For visualization purposes, we developed a dominance index ($\delta$). The dominance index for the sugar–acid pairs is written as $\delta_i = -|\varepsilon_{12}|$ when the sugar dominates and as $\delta_i = |\varepsilon_{12}|$ when the acid dominates. The dominance index for the sugar–sugar and acid–acid pairs is written as $\delta_i = -|\varepsilon_{12}|$ when the focal carbon source (glucose or succinate) dominates and as $\delta_i = |\varepsilon_{12}|$ when the additional carbon source dominates.

## Isolation of strains

Several communities (transfer 10) from different inocula and carbon sources were plated on chromogenic agar (HiCrome Universal differential Medium, Sigma) and grown for 48 hr at 30°C. Single colonies exhibiting distinct morphologies and/or colours were picked, streaked a second time on fresh chromogenic agar plates for purity, and grown for 48 hr at 30°C. A single colony was then picked from each plate and grown into Tryptic Soy Broth (TSB) for 48 hr at 30°C. The single-strain cultures were stored with 40% glycerol at −80°C. The isolated strains were sent for full-length 16S rRNA Sanger sequencing (Genewiz), and their taxonomy was assigned using the online RDP naïve Bayesian rRNA classifier version 2.11.

## Growth rate estimation

Twenty-two isolated strains belonging to the four dominant families, namely Enterobacteriaceae (7), Pseudomonadaceae (5), Moraxellaceae (6), and Rhizobiaceae (4) (*Supplementary file 1b*), were streaked from frozen stock on chromogenic agar plates and grown for 48 hr at 30°C. For each strain, a single colony was pre-cultured in 500 µL TSB in a deepwell plate for 24 hr at 30°C. Each strain was then acclimated into the 10 single carbon sources (glucose, fructose, cellobiose, ribose, glycerol, succinate, fumarate, benzoate, glutamine, and glycine). For this, 2 µL of the grown pre-culture was inoculated into 500 µL of fresh minimal media with each carbon source at a concentration of 0.07 C-mol/l and grown for 48 hr at 30°C. The growth curve assay was then performed in a 384-well plate by inoculating 1 µL of the grown isolate culture on 100 µL of fresh media of the same carbon source as for the acclimation step (three to four replicates each). OD620 was read every 10 min for ~40 hr at 30°C. The average growth rate of each strain in each carbon source was calculated as $r_{avg} = log_2(N_f/N_i)/(t_f-t_i)$ where $N_f$ is the OD at 16 hr (ie. $t_f$) and $N_i$ is the OD at 0.5 hr (i.e. $t_i$).

## Growth rate asymmetry calculation

The growth rate asymmetry on sugars ($q_S$) is calculated as $q_S = r_{avg}(E, S) - r_{avg}(O, S)$ where $r_{avg}(E, S)$ is the mean average growth rate of Enterobacteriaceae on the sugar $S$, and $r_{avg}(O, S)$ is the mean average growth rate of one of the other dominant families (i.e. Pseudomonadaceae, Moraxellaceae, or Rhizobiaceae) on $S$. The growth rate asymmetry on organic acids ($q_A$) is calculated as $q_A = r_{avg}(O, A) - r_{avg}(E, A)$ where $r_{avg}(O, A)$ is the mean average growth rate of one of the other dominant families (i.e. Pseudomonadaceae, Moraxellaceae, or Rhizobiaceae) on the organic acid $A$, and $r_{avg}(E, A)$ is the mean average growth rate of Enterobacteriaceae on $A$.

## Microbial consumer-resource model

Microbial community assembly is modeled using the Microbial Consumer Resource Model (MiCRM), with simulations implemented using *Community Simulator*, a freely available Python package (*Marsland et al., 2020b*). This model has been outlined extensively in previous work and has been shown to qualitatively reproduce ecological patterns across both natural (*Goldford et al., 2018*) and laboratory (*Marsland et al., 2020a*) microbiomes. Here we describe the exact equations simulated and parameters used in this paper. A more general description of this model is given elsewhere (*Marsland et al., 2019*; *Marsland et al., 2020b*). Our MiCRM simulations model the abundance $N_i$ of $n$ species and the abundance $R_\alpha$ of $M$ resources in a well-mixed chemostat-like ecosystem with continuous resource flow. We focus on continuous resource flow for simplicity and because previous work has shown that the major qualitative features of the MiCRM are unaffected by periodic resource supply (as was the case in our experiments) (*Marsland et al., 2020a*). Species interact by

uptake and release of resources into their environment. The dynamics of the system are governed by the following set of ordinary differential equations:

$$\frac{dN_i}{dt} = N_i \left[ \sum_\alpha (1-l) R_\alpha c_{i\alpha} - m \right] \tag{1}$$

$$\frac{dR_\alpha}{dt} = \frac{R_\alpha^0 - R_\alpha}{\tau} - \sum_j N_j R_\alpha c_{i\alpha} + \sum_{j,\beta} N_j D_{\alpha,\beta} R_\beta c_{j\beta} l \tag{2}$$

Here $c_{i\alpha}$ is the uptake rate of resource $\alpha$ by species $i$, $m$ is the minimal energy requirement for maintenance of species $i$, $\tau$ is the timescale for supply of external resources, $R_\alpha^0$ is the abundance of resource $\alpha$ supplied (i.e. the abundance in the media), $l$ is the fraction of resource secreted as by-product, and $D_{\alpha,\beta}$ is the fraction of resource $\alpha$ secreted as by-product $\beta$. In line with previous work, the following parameters are kept constant for all simulations $\tau = 1$, $m = 1$, and $l = 0.5$ (*Goldford et al., 2018*; *Marsland et al., 2020a*).

In the MiCRM, by-product production is encoded in the metabolic matrix $D$ where each element $D_{\alpha,\beta}$ specifies the fraction of resource $\alpha$ secreted as by-product $\beta$. As in previous work, each column $\beta$ in $D_{\alpha,\beta}$ is sampled from a Dirichlet distribution with concentration parameter $d_{\alpha,\beta}=1/(sM)$ where $s = 0.3$ is a parameter that tunes the sparsity of the underlying metabolic network. The Dirichlet distribution ensures that each column sums to one so that the total secretion flux does not exceed the input flux. For simplicity we used a fixed concentration parameter and so are not assuming any underlying metabolic structure. The MiCRM also assumes that all species have the same $D$ matrix, that is when growing on the same resource each species releases the same metabolic by-products.

In our simulations, species differ solely in the uptake rate for different resources $c_{i\alpha}$ where $i$ is the species and $\alpha$ is the resource. Taxonomic specialization is introduced in the form of two families $F_A$ and $F_S$ that each have a preference for one of two resource classes $A$ and $S$, respectively. Each $c_{i\alpha}$ is sampled from a gamma distribution (to ensure positivity) whose mean $<c_{i\alpha}>$ and variance var($c_{i\alpha}$) depends on the family of $i$ and the resource class of $\alpha$. This means that all species are capable of metabolizing all resources. Specifically

$$\langle c_{i\alpha} \rangle = \begin{cases} \frac{\mu_c}{M}(1+q_A) \text{ if } i \in F_A \text{ and } \alpha \in A \\ \frac{\mu_c}{M}(1-q_A) \text{ if } i \in F_S \text{ and } \alpha \in A \\ \frac{\mu_c}{M}(1+q_S) \text{ if } i \in F_S \text{ and } \alpha \in S \\ \frac{\mu_c}{M}(1-q_S) \text{ if } i \in F_A \text{ and } \alpha \in S \end{cases}$$

and

$$var(c_{i\alpha}) = \begin{cases} \frac{\sigma_c^2}{M}(1+q_A) \text{ if } i \in F_A \text{ and } \alpha \in A \\ \frac{\sigma_c^2}{M}(1-q_A) \text{ if } i \in F_S \text{ and } \alpha \in A \\ \frac{\sigma_c^2}{M}(1+q_S) \text{ if } i \in F_S \text{ and } \alpha \in S \\ \frac{\sigma_c^2}{M}(1-q_S) \text{ if } i \in F_A \text{ and } \alpha \in S \end{cases}$$

$\mu_c = 10$ determines the overall mean uptake rate and $\sigma_c^2 = 3$ determines overall variance in uptake rate (these parameters are the default value in the *Community Simulator* package). Parameters $q_A$ and $q_S$ tune the relative advantage each specialist family has on its preferred resource. When $q_A = 1$, only $F_A$ consumes resources in $A$ whereas when $q_A = 0$ both families have equal access to resources in $A$. Conversely, when $q_S = 1$, only $F_S$ consumes resources in $S$ whereas when $q_S = 0$ both families have equal access to resources in $S$.

For each simulation we consider 200 species (100 per family). Each community in one simulation is seeded with all 200 species. This means that there is no stochasticity in colonization (though see *Figure 4—figure supplement 2* where this assumption is relaxed). We choose 200 species as this is within the range of the number of ESVs in a typical inoculum for our experiments (110–1290 ESVs, reported in *Goldford et al., 2018*). The initial abundances are all set to 1 for simplicity. In line with our experiments, either one or two resources are supplied in the media and the rest are generated as metabolic by-products. For simulations with a single supplied resource, $R_\alpha^0 = 1000$ if $\alpha$ is the supplied resource and 0 otherwise. For simulations with two supplied resources, $R_\alpha^0 = 500$ for each

supplied resource and 0 otherwise. This ensures that the total amount of resources is kept constant as in our experiments. In total, we consider 20 resources in each simulation (with 10 resources in each resource class [*A* or *S*]) as this gives us communities with $7 \pm 2$ species (mean ± SD) at equilibrium, which is comparable to the diversity of our experimental communities.

In line with our experiments, each simulation consisted of three types of mixed-resource environments (one with two supplied resources in class $R_A$, one with two supplied resources in class $R_S$ and one with one resource in class $R_A$ and one resource in class $R_S$). We also included all four single resource environments needed to predict the mixes (i.e. two with the resources in class $R_A$ and two with the resources in class $R_S$). Therefore, each simulation consisted of seven communities each in a different environment and all seeded with the same initial set of 200 species. The equilibrium for all seven communities was found using the SteadyState function in *Community Simulator* (*Marsland et al., 2020b*). Failed runs where the SteadyState function returned an error were removed from our analysis. In addition, for each simulation we tested that the SteadyState algorithm had truly converged to an equilibrium using the same approach as in *Marsland et al., 2020a* and removed all non-convergent runs (defined as a run for which $|d \ln (N_i / dt)| > 10^{-5}$). Including these runs would not have qualitatively changed our results.

In the raw numerical output of the run, all species have non-zero abundances due to limits in numerical precision. A species was considered extinct if its abundance was less than $10^{-6}$, which was set by looking at a histogram of the raw output of our simulations. Once the extinct species were removed, we predicted the relative abundance of each species *i* in the mixture of nutrients using the same approach that had been used for the experimental data. To obtain a statistically robust sample size, we repeated this procedure for 100 replicate simulations, resampling all randomly generated parameters in each simulation (i.e. resampling all $c_{i\alpha}$ and $D_{\alpha'\beta}$ as described above).

## Flux balance analysis

We use FBA to estimate whether the different carbon sources were likely to result in large differences in oxygen demand. FBA is a widely used constraint-based modelling approach that allows us to predict metabolic fluxes through a stoichiometric metabolic network (assuming optimal growth and that cells are in a steady state) (*Orth et al., 2010*). For this analysis, we used a modified version of iJO1366 (see below), a high-quality genome scale-metabolic network of *E. coli* (*Orth et al., 2011*). FBA simulations were performed using the COBRApy Package (*Ebrahim et al., 2013*). We simulated the growth of *E. coli* on minimal synthetic media in aerobic conditions containing one of the 10 carbon sources used in our experiments (*Figure 2A*). These simulations were used to estimate the number of $O_2$ molecules that would be consumed per carbon atom when growing on each of the 10 carbon sources (*Figure 4—figure supplement 5*). Except for benzoate, we found that all of the carbon sources exhibited similar predicted oxygen demands (0.25–0.34 $O_2$/C). This does not rule out the possibility that the kinetics of growth and $O_2$ uptake may contribute to increased $O_2$ depletion in one carbon source compared to another, nor that the different taxa selected for by the different carbon sources might display differences in $O_2$ uptake. Nonetheless, they do suggest that differences in oxygen stoichiometry are unlikely to be the main mechanism for dominance across all carbon sources.

For these simulations, all inorganic compounds were assumed to be in excess and their exchange fluxes were unbounded by setting to an arbitrarily large negative value (−1000 mmol/gDWh). These compounds are as follows: ca2_e, cbl1_e, cl_e, co2_e, cobalt2_e, cu2_e, fe2_e, fe3_e, h_e, h2o_e, k_e, mg2_e, mn2_e, mobd_e, na1_e, nh4_e, ni2_e, pi_e, sel_e, slnt_e, so4_e, tungs_e, zn2_e, and o2_e. To estimate the optimal oxygen consumption per mole of carbon consumed, the exchange flux for each of the 10 carbon sources was set to −1 cmol/gDWh. We set the lower bound on ATPM maintenance to 0 as we wanted to estimate the $O_2$/C when resources were in excess and so the effects of growth independent maintenance would be negligible. Similar results can be obtained using the default ATPM lower bound in the published model and setting a higher lower bound on the carbon uptake flux (such as the −60 cmol/gDWh typically used for *E. coli* on glucose) (*Harcombe et al., 2014*). The biomass reaction (BIOMASS_Ec_iJO1366_core_53p95M) was used as the objective function.

In the absence of any additional reactions, FBA predicts that *E. coli* can grow on all carbon sources aside from benzoate and cellobiose. To obtain growth on cellobiose we added a cellobiose PTS transporter (CELBpts) and a beta-glucosidase (BGLA1) as well the corresponding exchange and

periplasm diffusion reactions (EX_cellb_e and CLBtex). While most wild-type *E. coli* are unable to use cellobiose, other Enterobacteriaceae such as *Klebsiella oxytoca* can (using the reactions we have added) (*Sekar et al., 2012*). To obtain growth on benzoate, we added the benzoate degradation pathway (the β-ketoadipate pathway) from *Pseudomonas putida*. This included the following reactions (EX_bz_e, BZt, BZ12DOX, CATDOX, MUCCYCI, MUCLI, OXOAEL, 3OADPCOAT) (*Sudarsan et al., 2016*). We note that the lack of growth by *E. coli* on benzoate is consistent with the low abundance of other Enterobacteriaceae in the benzoate communities and the prevalence of *Pseudomonas* (*Figure 2—figure supplement 1*). The $O_2$/C ratio obtained on benzoate using the *E. coli* model (0.560) is similar to the ratio obtained using the *P. putida* iJN1463 model (0.548) in which the benzoate degradation pathway is native (*Nogales et al., 2020*).

## Acknowledgements
We want to thank members of the Sanchez lab for helpful discussions. This work was supported by the National Institutes of Health through grant 1R35 GM133467-01, and by a Packard Foundation Fellowship to AS.

## Additional information

### Funding

| Funder | Grant reference number | Author |
|---|---|---|
| National Institutes of Health | 1R35 GM133467-01 | Alvaro Sanchez |
| David and Lucile Packard Foundation | | Alvaro Sanchez |

The funders had no role in study design, data collection and interpretation, or the decision to submit the work for publication.

### Author contributions
Sylvie Estrela, Conceptualization, Data curation, Formal analysis, Investigation, Visualization, Writing - original draft, Writing - review and editing; Alicia Sanchez-Gorostiaga, Conceptualization, Investigation, Writing - review and editing; Jean CC Vila, Conceptualization, Formal analysis, Investigation, Methodology, Writing - review and editing; Alvaro Sanchez, Conceptualization, Supervision, Funding acquisition, Writing - review and editing

### Author ORCIDs
Sylvie Estrela  https://orcid.org/0000-0003-1946-3657
Alicia Sanchez-Gorostiaga  https://orcid.org/0000-0002-2719-5659
Jean CC Vila  https://orcid.org/0000-0003-0499-0200
Alvaro Sanchez  https://orcid.org/0000-0002-2292-5608

### Decision letter and Author response
Decision letter https://doi.org/10.7554/eLife.65948.sa1
Author response https://doi.org/10.7554/eLife.65948.sa2

## Additional files

### Supplementary files
• Supplementary file 1. Supplementary tables. (**a**) Carbon sources used in this study. (**b**) Taxonomy of strains used in the growth rate assay and community they were isolated from.

• Transparent reporting form

### Data availability

The 16S rRNA sequencing abundance data and community OD are available at Dryad (https://doi.org/10.5061/dryad.3bk3j9kj6) The raw 16S rRNA amplicon sequences and metadata file have been deposited in the NCBI SRA database under Project ID PRJNA723201. The source data files and scripts used to generate the figures are available at https://github.com/sylestrela/Estrelaetal2021_NutrientDominance (copy archived at https://archive.softwareheritage.org/swh:1:rev:4c12cbaee6c-fa187b2abd2b816a7e5b8103d8f88/) and the code for the CRM and FBA simulations are available at https://github.com/vilacelestin/Estrelaetal2021 (copy archived at https://archive.softwareheritage.org/swh:1:rev:f97ad5689defe97f02800f6dd1825d1eea3bdc61/).

The following dataset was generated:

| Author(s) | Year | Dataset title | Dataset URL | Database and Identifier |
|---|---|---|---|---|
| Estrela S, Sanchez-Gorostiaga A, Vila JCC, Sanchez A | 2021 | Data from: Nutrient dominance governs the assembly of microbial communities in mixed nutrient environments | http://dx.doi.org/10.5061/dryad.3bk3j9kj6 | Dryad Digital Repository, 10.5061/dryad.3bk3j9kj6 |

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
