## [Decision Letter]

**Acceptance summary:**

This manuscript is of broad interest to readers in the field of microbial ecology and systems biology, especially for researchers who study the assembly of multi-species communities. The authors examine how the presence of one or two different carbon sources affect the growth of microbial communities within the corresponding environment. The work combines mathematical modeling, experiments profiling the taxonomic composition of model communities grown on different carbon source(s), and measurements of the growth rate of isolates from these communities to arrive at the conclusion that in some contexts, one carbon source has a dominant effect on the composition of a community.

**Decision letter after peer review:**

Thank you for submitting your article "Nutrient dominance governs the assembly of microbial communities in mixed nutrient environments" for consideration by *eLife*. Your article has been reviewed by 3 peer reviewers, and the evaluation has been overseen by a Reviewing Editor and Meredith Schuman as the Senior Editor. The following individuals involved in review of your submission have agreed to reveal their identity: Silvio Waschina (Reviewer #1); Babak Momeni (Reviewer #2).

Essential Revisions:

1. Statistical analyses: There are several issues regarding the statistical analysis of the data.

a. One concern is about the statistical analysis of abundances across different cases (e.g. the results in Figure 2C). The authors have averaged the deviation in abundances (epsilon) across different cases. However, this is not a fair comparison. For example, a change from 1 to 11% in abundance is not comparable to a change from 70 to 80%. I recognize that alternative approaches have limitations as well (e.g. noise if the change in abundance is normalized to abundance). Nevertheless, the authors should avoid basing their conclusions (e.g. lines 187-191) on the mean of epsilon, because those conclusions will be biased by cases that have higher abundance.

b. The authors use one-sample t-tests to verify whether the growth observed in the two carbon source environments differs significantly from the expected growth. However, this test assumes that there is no variation in the predicted values. Since this assumption is clearly wrong, the authors should use a two-sample t-test (or any other equivalent) instead.

2. There are several cases of dominance (one genus/family taking over the community) when specific sugars are supplied. This observation raises the question whether these cases can skew the conclusions. For example, the null prediction will hold if there are extreme specialists that utilize preferentially one particular resource. However, what happens if only cases without dominance by one genus/family were included in the analysis? The authors should briefly discuss whether and how this situation would affect their results.

3. It would be helpful to have information about growth yield on the different carbon sources, in addition to growth rate. Thus, if possible (no additional experiment required), it would be good to include OD measurements or DNA concentrations from the extractions to provide information on the yield of communities on different amounts of a single carbon source.

4. In Figure S9, it looks like there is poor growth overall on benzoate and glycine. On glycine in particular, the fastest growth rates that are reported seem to be close to 0.01/h, or one doubling every 100 hours. In the community assembly experiments, communities are transferred every 48 h with a dilution rate of 125x. Given these slow growth rates, it remains unclear how these communities could have survived on these two carbon sources for 10 serial passages. Is it possible that the growth rates differ between the 384 well plate and the 96-well deep well plates? Or were the isolates sampled not representative of members of the complex community in general? Did the communities as a whole grow on glycine and benzoate? Please clarify this issue.

5. It remains generally unclear whether or not the experimental system is carbon limited across all of the different conditions and in all resource combinations tested. If growth on one carbon source requires a lot of oxygen consumption, and growth on the other carbon source requires less, then oxygen may be a hidden limiting resource. This type of dynamic could provide an additional reason why pairs of similar nutrients are better predicted than pairs of dissimilar nutrients: similar nutrients enter the central metabolism at similar points and are more likely to consume additional resources in a stoichiometrically similar manner. The authors should provide evidence that rules out this possibility.

6. Given that the consumer resource model is a valuable addition to the manuscript, it would be better to move it to the Results section. Moreover, a more in-depth discussion of the role specific resources play for metabolism and growth should be included. In particular, when presenting and discussing Figure 4, it would be helpful to briefly state how the species and families (in terms of their nutrient preference) were set up in the model. In this context, the authors also may want to consider mentioning some of the work by Hwa lab and Egli lab, as they may shed some light on the underlying mechanisms of the observed patterns in this manuscript.

Reviewer #1 (Recommendations for the authors):

line 52 (abstract): "sugars generally dominate organic acids." From the context it is not immediately clear what is meant by "dominate". The meaning becomes clear when reading the manuscript, but the meaning of "dominance" remains vague when reading the abstract only.

line 74 & line 81: Please briefly explain the term "enrichment community" as this might not be directly clear to all the potential readers.

Lines 107-110: The construction of the null expectation model is well-explained and justified. Yet, I was surprised by the sentence about the potential ecological and metabolic interactions between species (lines 107-110), which states that these are not affected by mixing nutrients. For instance, in the prominent example of overflow metabolism some bacterial species do not oxidize sugars completely but excrete notable amounts of organic acids (often acetate), which can serve as nutrient for acetate-oxidizing bacteria. In such a situation, the metabolic interaction would be strongly influenced by the combination of available nutrients; say two sugars enable acetate cross-feeding while a combination of two organic acids could potentially disable the interactions with significant impact on community assembly. The mentioned sentence (lines 107ff) gives the impression that nutrient combinations that interactively influence community assembly by altering metabolic interactions between species cannot be detected as deviation of the observed community structure to the expected structure based on the null (additive) model. In short, my suggestions would be to differentiate in the text between nutrient-independent metabolic interactions (i.e. those interactions taken place independently of the available nutrient(s)) and nutrient combination-dependent metabolic interactions (i.e. interactions that are affected in their expression/flux based on the identity of nutrient combinations).

lines 139-140: Although you specify this in the methods section, it would be nice if you could add to this sentence on which basis the 1:1 carbon source ratio was calculated to prevent confusion between molar, mass and C-molar ratio. Also, it would be great to briefly explain the motivation why carbon source concentrations were adjusted to equal C-molar concentrations (and not mass or molar) in the methods section and to state the final molar concentrations for each respective single CS media in Table S1.

lines 145-148: Are there, besides to the diversity in biochemistry, specific reasons for the choice of the ten carbon sources? Are these also relevant/potential nutrients for soil microorganisms?

line 150: This is the first occurrence of the abbreviation of ESV. Please state the full term here.

Reviewer #2 (Recommendations for the authors):

The authors in this manuscript examine the impact of the metabolites on the composition of microbial communities. Specifically, they search for a link between the composition of a community when supplemented with a combination of carbon sources with communities formed under each of those sources. To address this, they examine the assembly of communities in defined nutrient mixes through enrichment.

In my opinion, the setup and approach they have chosen makes perfect sense for addressing the question they have posed. Overall, I think this manuscript is a helpful step forward to explain some of the patterns that have been observed in microbiome studies.

1. I have a concern about the statistical analysis of abundances across different cases. This applies, for example, when interpreting the results in Figure 2C. The authors have averaged the deviation in abundances (epsilon) across different cases. However, I am not sure if this offers a fair comparison. For example a change from 1 to 11% in abundance is not comparable to a change from 70 to 80%. I recognize that alternative approaches have limitations as well (e.g. noise if the change in abundance is normalized to abundance). Nevertheless, I would discourage basing the conclusions (e.g. lines 187-191) on the mean of epsilon, because those conclusions will be biased by cases that have higher abundance.

Reviewer #3 (Recommendations for the authors):

1. It would be helpful to have information about growth yield on the different carbon sources, in addition to growth rate. Maybe you measured OD prior to collecting samples prior to DNA extraction? Or maybe you have the DNA concentrations from the extractions? Also, if there is any information on the yield of communities on different amounts of a single carbon source, that would be excellent to include. This is not to suggest more experiments!

2. In Figure S9, it looks like there is poor growth overall on benzoate and glycine. On glycine in particular, the fastest growth rates that you are reporting seems close to 0.01/h, or one doubling every 100 hours. In your community assembly experiments, you transfer every 48 h with a dilution rate of 125x. After 10 passages, I don't understand how a community made of strains growing at these slow rates could survive on these two carbon sources. Is it possible that the growth rates between the 384 well plate and the 96-well deep well plates differ? Or were the isolates sampled not representative of members of the complex community in general? Did the communities as a whole grow on glycine and benzoate?

3. I would like more evidence to support the assertion that the experimental system is carbon limited across all of the different conditions and in all resource combinations. If growth on one carbon source requires a lot of oxygen consumption, and growth on the other carbon source requires less, then oxygen may be a hidden limiting resource. This type of dynamic could provide an additional reason why pairs of similar nutrients are better predicted than pairs of dissimilar nutrients- similar nutrients enter central metabolism at similar points and are more likely to consume additional resources in a stoichiometrically similar manner.

[Editors' note: further revisions were suggested prior to acceptance, as described below.]

Thank you for submitting your article "Nutrient dominance governs the assembly of microbial communities in mixed nutrient environments" for consideration by *eLife*. The evaluation has been overseen by Christian Kost (Reviewing Editor) and Meredith Schuman (Senior Editor).

We think that the revised version of the manuscript has significantly improved in terms of clarity. Also, you have sufficiently addressed all comments raised by the reviewers. However, there are two points left we would like you to revise before we can finally accept the manuscript for publication in *eLife*:

1. In the current version, you included the analysis on the O2-requirement for utilising the different carbon sources in the Discussion section. Please report this result in the Results section.

2. The panels of the figures are inconsistently labelled (i.e. the way panels A,B,C are presented). Please arrange these panels consistently in all figures from left to right and top to bottom in case panels fill more than one row.

---

## [Author Response]

Essential Revisions:1. Statistical analyses: There are several issues regarding the statistical analysis of the data.a. One concern is about the statistical analysis of abundances across different cases (e.g. the results in Figure 2C). The authors have averaged the deviation in abundances (epsilon) across different cases. However, this is not a fair comparison. For example, a change from 1 to 11% in abundance is not comparable to a change from 70 to 80%. I recognize that alternative approaches have limitations as well (e.g. noise if the change in abundance is normalized to abundance). Nevertheless, the authors should avoid basing their conclusions (e.g. lines 187-191) on the mean of epsilon, because those conclusions will be biased by cases that have higher abundance.

We agree that averaging epsilon across different cases is not a fair comparison here, and we wish to thank the editor and reviewers for pointing this out. In light of this comment and comment (1b) below about the type of t-test used, we repeated our statistical analyses to re-calculate epsilon and its statistical significance for each carbon source pair and family. This is now described in the Materials and methods section (lines 536-552) and reads as follow:

“To determine whether an interaction between nutrients exists (i.e. ε ≠ f_i,12_- f_i,12_(null)), we assess whether the abundance observed in the carbon source mixture is significantly greater or lower than the abundance predicted by the null additive model (i.e. ε > 0 or ε < 0, respectively) (onetailed paired t-test). More specifically, considering the two inocula and four replicates per carbon source, the family-level analysis was done as follow. For each carbon source pair and inoculum, four predicted pairs are formed by randomly pairing one replicate of each carbon source. Four unique observed vs predicted pairs are then randomly formed from the 64 possible combinations (i.e. from the four single nutrient 1 x four single nutrient 2 x four mixed nutrients 12). Up to this point, the pairs are formed for each inoculum separately, in other words, there is no cross-inocula pairing. Once all N=8 pairs are formed (i.e. N=4 pairs per inoculum), they are pooled to perform the one-tailed paired t-test. The N=8 pairs are then randomly permuted 1000 times, determining the t-statistic for each permutation. We establish a 95% confidence threshold for the t-statistic. The effect observed is statistically significant (i.e. an interaction exists) if a significant difference is found in more than 95% of the permuted pairs. At the genus-level, the analysis was performed in a similar way, except that the two inocula were kept separately. This is because, compared to families, the likelihood that genera that are sampled in one of the inocula are sampled in the other inoculum is much lower.”

Using this statistical analysis approach, we overall find similar qualitative results (updated Figure 3C and Figure 3—figure supplement 1, Figure 3—figure supplement 2, Figure 3—figure supplement 3, Figure 3—figure supplement 4). There is a small increase in the fraction of ‘no interaction’ cases at the family-level, and a greater increase at the genus-level (which might also be magnified by the lack of statistical power).

We have also now rephrased the sentence lines 187-191 (now lines 195-199) as:

“For example, across all succinate-sugar pairs, Enterobacteriaceae are significantly more abundant than predicted by the null model (one-tailed paired t-test, N=8, p<0.05 based on 1000 permutations, see Methods) while both Rhizobiaceae and Moraxellaceae are less abundant than predicted (one-tailed paired t-test, N=8, p<0.05 based on 1000 permutations, see Methods) (Figure 2C).”

b. The authors use one-sample t-tests to verify whether the growth observed in the two carbon source environment differs significantly from the expected growth. However, this test assumes that there is no variation in the predicted values. Since this assumption is clearly wrong, the authors should use a two-sample t-test (or any other equivalent) instead.

The reviewer is correct, thank you for pointing this out. We have now repeated our statistical analysis using a one-tailed paired t-test (see response to comment 1a above).

2. There are several cases of dominance (one genus/family taking over the community) when specific sugars are supplied. This observation raises the question whether these cases can skew the conclusions. For example, the null prediction will hold if there are extreme specialists that utilize preferentially one particular resource. However, what happens if only cases without dominance by one genus/family were included in the analysis? The authors should briefly discuss whether and how this situation would affect their results.

We thank the reviewer for this interesting comment, and we are grateful for the opportunity to clarify this point. As suggested by the reviewer, we also expect our null model to be less predictive if all species/families are equally good at growing on all carbon sources. The reason why we did not consider such ‘generalist’ scenario is because in recent work with enrichment communities from our lab, we have found evidence for strong carbon source preference and specialization (Estrela et al. 2020). For instance, we find that Enterobacteriaceae are glucose specialists (meaning by this that they grow on average 60% faster than the Pseudomonadaceae) while *Pseudomonas* are organic acid specialists (meaning by that they grow, on average 46% faster than the Enterobacteriaceae). Inspired by this comment, in the revised version we now use our CRM to explore how the degree of resource specialization can affect the predictive power of our null model. We find that as the level of specialization decreases (i.e. species become more generalists), the model becomes less good at predicting the mixture composition from the single nutrient composition. We have now added a Supplementary Figure (Figure 4—figure supplement 3) and the following sentence to the main text (lines 337-339)

“The predictive accuracy of our null model is, however, influenced by the level of resource specialization. The less specialized (i.e. more generalist) the families are, the lower the predictive power of the null additive model (Figure 4—figure supplement 3).”

3. It would be helpful to have information about growth yield on the different carbon sources, in addition to growth rate. Thus, if possible (no additional experiment required), it would be good to include OD measurements or DNA concentrations from the extractions to provide information on the yield of communities on different amounts of a single carbon source.

The OD measurements are part of the calculation of the predicted abundances and are included in the dataset. For clarity, we have now added a new figure (Figure 2—figure supplement 5) with the OD of the communities in each single carbon source.

4. In Figure S9, it looks like there is poor growth overall on benzoate and glycine. On glycine in particular, the fastest growth rates that are reported seem to be close to 0.01/h, or one doubling every 100 hours. In the community assembly experiments, communities are transferred every 48 h with a dilution rate of 125x. Given these slow growth rates, it remains unclear how these communities could have survived on these two carbon sources for 10 serial passages. Is it possible that the growth rates differe between the 384 well plate and the 96-well deep well plates? Or were the isolates sampled not representative of members of the complex community in general? Did the communities as a whole grow on glycine and benzoate? Please clarify this issue.

We thank the reviewer for the opportunity to clarify the confusion around Figure S9. We would like to highlight the following points:

a. We want to emphasize that the reason we measured these growth curves was to estimate whether the competitive ability of specialists in different resources exhibited an asymmetry. The average growth rates shown in Figure S9 are for a collection of strains that were isolated from a few of the communities (listed in the Table in the Supplementary file 1B). Two strains (both *Pseudomonas*) were isolated from the benzoate communities and none of the strains were isolated from the glycine communities. As we now show in the new Figure 2—figure supplement 5, the communities do grow on benzoate and on glycine. As we show in the plot in Author response image 1, at least one strain from most families was able to grow to an OD>0.1 in benzoate and glycine too.

b. Most importantly, what we show in Figure S9 is the average growth rate over the first 16 hours (from t=0.5h to t=16h), rather than the maximum growth rate attained over the entire 48 hour incubation. The reason why we focus on the average growth rate during the first 16 hours as a proxy for competitive ability is two-fold. First, the average growth rate takes into account both the duration of the lag phase and the maximum growth rate, and therefore it is a better descriptor of the competitive ability of a microbe in the supplied substrate than the maximum growth rate alone. To see why this is the case, consider for instance the competition between a strain with a very fast maximum growth rate but a very long lag-phase, and a second strain with a somewhat slower maximum growth rate but no lag-phase. By the time the former escapes its lag-phase, the latter may have already depleted the resource. If, on the other hand, the second strain grows extremely slowly, it will be unable to take advantage of its short lag-phase. We reasoned that since the average growth rate over the first few hours of growth (we have previously found that 16 h is the typical time required for the depletion of the supplied resource) incorporates both the growth rate and the lag into a single parameter, it represents a better descriptor of the competitive growth ability of a strain. Second, and along the same lines, we use the first 16h of growth rather than a longer time window because we want to determine the growth rate on the supplied resource and avoid potential artifacts from growth on secretions. Thus, strains with a long lag and/or a slow growth rate will have a small average growth rate, yet they may reach a reasonably high OD at the end of the 48h incubation period. This is the case for some of the strains sampled. In Author response image 1 we show the 48h growth curves of each strain on benzoate and glycine. We can see that some of the *Pseudomonas* strains (and to a lesser extent other strains as well) grow well on benzoate and on glycine. In both cases, multiple strains reach high ODs over the total period of 48 hours, even when the average growth rate over the first 16 hours is slow.

**Author response image 1. sa2fig1:** 

To clarify this point, we have now added the following sentence to the legend of Figure S9 (now labelled as Figure 4—figure supplement 1):“Thus, this approach takes into account both lag and growth rate, two growth traits that are important in determining the competitive ability of a strain. We use the first 16h of growth rather than a longer time window to better assess growth rate on the supplied nutrient and avoid potential artifacts from growth on secretions.”

5. It remains generally unclear whether or not the experimental system is carbon limited across all of the different conditions and in all resource combinations tested. If growth on one carbon source requires a lot of oxygen consumption, and growth on the other carbon source requires less, then oxygen may be a hidden limiting resource. This type of dynamic could provide an additional reason why pairs of similar nutrients are better predicted than pairs of dissimilar nutrients: similar nutrients enter the central metabolism at similar points and are more likely to consume additional resources in a stoichiometrically similar manner. The authors should provide evidence that rules out this possibility.

We appreciate the point raised by the reviewers and we completely agree that O2 limitation can be a potential mechanism of nutrient dominance. Other mechanisms exist as well that could generate dominance. For instance, one of the nutrients may be toxic to some strains (but not to those it selects), or its uptake may be inhibited when the other nutrient is present. The point we wanted to make through our Consumer-Resource Model is to show that dominance is a general outcome of consumer-resource interactions when there exists an asymmetry in nutrient value, and that such asymmetry is indeed observed for the families that we are finding in our communities. In other words, even when other more specific mechanisms are absent, dominance may still be easily observed. However, other mechanisms may be also at play too and if anything, the wealth of independent mechanisms that may contribute to dominance illustrates the potential importance of this phenomenon.

With regard to O2 in particular, the reviewer’s comment prompted us to use flux-balance analysis (FBA) to determine the O2 demands of different carbon sources. We found that all of them are similar (new Figure 4—figure supplement 5) with the exception of benzoate, which exhibits a higher oxygen demand. However, this does not rule out the possibility that kinetics of growth and O2 uptake may still contribute to O2 depletion in a manner that may further stimulate dominance (in addition to the asymmetric resource benefits we report in Figure 4). As we discuss in comment (6) below, other mechanisms such as synergistic co-utilization, diauxie, or selective inhibition may contribute to dominance as well. Identifying all of the mechanisms at play for every family and pair of nutrients in our sample would be a gargantuan effort that is well beyond what we can accomplish in this paper. For that reason, we have been cautious in our claims and this was also a reason why we had originally placed the model in the Discussion. To more clearly emphasize this point, we have now added a paragraph in the Discussion where we more explicitly discuss potential additional mechanisms that may lead to dominance in our communities beyond the resource consumption asymmetry (lines 405-443). It now reads:

“It is important to recognize, however, that other explanations and mechanisms of dominance may be at play too. Generally, these can be split into two main categories: asymmetries in how species respond to the provided nutrient and asymmetries that emerge as a result of the constructed environment. Below, we discuss several specific mechanisms that may contribute to each of these. […] Importantly, even if species respond to the supplied pair of nutrients in an additive manner, niche construction (and thus the interactions between species) may not be additive. For example, species may secrete secondary metabolites or antimicrobial agents on nutrient mixtures, which may interact with each other [33-35]. Moreover, cellular growth can change other physico-chemical properties of the environment aside from carbon source availability (such as by changing the pH, or the accessibility of non-carbon source nutrients leading to co-limitation) [36-38]. For instance, one plausible mechanism that could lead to dominance is oxygen limitation, in particular if different carbon sources were to have different oxygen requirements [39-40]. To explore this idea of asymmetric oxygen demands, we used flux-balance analysis (FBA) to determine the oxygen demands of growth on each of the single carbon sources. We found that, except for benzoate, all carbon sources have similar oxygen demands (Figure 4—figure supplement 5). This does not rule out, however, the possibility that kinetics of growth and oxygen uptake may still contribute to oxygen depletion in a manner that may further stimulate dominance (in addition to the asymmetric resource benefits we report in Figure 4).”

6. Given that the consumer resource model is a valuable addition to the manuscript, it would be better to move it to the Results section. Moreover, a more in-depth discussion of the role specific resources play for metabolism and growth should be included. In particular, when presenting and discussing Figure 4, it would be helpful to briefly state how the species and families (in terms of their nutrient preference) were set up in the model. In this context, the authors also may want to consider mentioning some of the work by Hwa lab and Egli lab, as they may shed some light on the underlying mechanisms of the observed patterns in this manuscript.

We are glad that the reviewer found the consumer-resource model a valuable addition to the manuscript. As we explained above (point 5), we initially presented the model in the Discussion section out of caution. We are aware that our model does not demonstrate the mechanism behind the observed dominance patterns, but rather it shows that the simplest consumer- resource model (which had already explained a lot of our observations in enrichment communities) naturally exhibits dominance under conditions of resource utilization asymmetry that seem to be consistent to the ones we find in our experiments. This does not rule out contributions from other potential mechanisms together with (or even instead of) this resource asymmetry in individual cases. Other potential mechanisms include: synergistic co-utilization, diauxie, and selective inhibition (e.g. antibacterial effect by one of the nutrients) (as discussed above and now in the Discussion lines 395-431, where we mention some of the work by the Hwa and Egli labs).

In response to the reviewer’s comments, we have now moved the model to the Results section and have also added the following sentences to briefly describe how nutrient preference was implemented in the model:

“Members of the family specializing on sugars (i.e. the Enterobacteriaceae) have on average a higher uptake rate on each sugar whereas members of the family specializing on acids (i.e. the Pseudomonadaceae) have on average a higher uptake rate on each acid. The magnitude of specialization by each family on its preferred resource type is tuned by two parameters, *q_A_* and 𝑞_S_, which modulate the mean and variance of the probability distribution from which the uptake rates are sampled (see Materials and methods for more details).”

[Editors' note: further revisions were suggested prior to acceptance, as described below.]

We think that the revised version of the manuscript has significantly improved in terms of clarity. Also, you have sufficiently addressed all comments raised by the reviewers. However, there are two points left we would like you to revise before we can finally accept the manuscript for publication in eLife:1. In the current version, you included the analysis on the O2-requirement for utilising the different carbon sources in the Discussion section. Please report this result in the Results section.

We have moved the analysis on the O2-requirement for utilizing the different carbon sources from the Discussion to the Results section.

2. The panels of the figures are inconsistently labelled (i.e. the way panels A,B,C are presented). Please arrange these panels consistently in all figures from left to right and top to bottom in case panels fill more than one row.

We have arranged the figure panels accordingly (revised Figure 4).